# Retinoic Acid Receptor Is a Novel Therapeutic Target for Postoperative Cognitive Dysfunction

**DOI:** 10.3390/pharmaceutics15092311

**Published:** 2023-09-13

**Authors:** Yongjie Bao, Wenni Rong, An Zhu, Yuan Chen, Huiyue Chen, Yirui Hong, Jingyang Le, Qiyao Wang, C. Benjamin Naman, Zhipeng Xu, Lin Liu, Wei Cui, Xiang Wu

**Affiliations:** 1Department of Anesthesiology, The First Affiliated Hospital of Ningbo University, Ningbo 315010, China; 196002770@nbu.edu.cn (Y.B.);; 2Translational Medicine Center of Pain, Emotion and Cognition, Ningbo Key Laboratory of Behavioral Neuroscience, Zhejiang Provincial Key Laboratory of Pathophysiology, School of Medicine, Ningbo University, Ningbo 315211, China; 3Li Dak Sum Yip Yio Chin Kenneth Li Marine Biopharmaceutical Research Center, Ningbo University, Ningbo 315211, China

**Keywords:** bioinformatics model, retinoic acid receptor, connectivity map, postoperative cognitive dysfunction, acitretin

## Abstract

Postoperative cognitive dysfunction (POCD) is a clinical syndrome characterizing by cognitive impairments in the elderly after surgery. There is limited effective treatment available or clear pathological mechanisms known for this syndrome. In this study, a Connectivity Map (CMap) bioinformatics model of POCD was established by using differently expressed landmark genes in the serum samples of POCD and non-POCD patients from the only human transcriptome study. The predictability and reliability of this model were further supported by the positive CMap scores of known POCD inducers and the negative CMap scores of anti-POCD drug candidates. Most retinoic acid receptor (RAR) agonists were negatively associated with POCD in this CMap model, suggesting that RAR might be a novel target for POCD. Most importantly, acitretin, a clinically used RAR agonist, significantly inhibited surgery-induced cognitive impairments and prevented the reduction in RARα and RARα-target genes in the hippocampal regions of aged mice. The study denotes a reliable CMap bioinformatics model of POCD for future use and establishes that RAR is a novel therapeutic target for treating this clinical syndrome.

## 1. Introduction

Postoperative cognitive dysfunction (POCD) is a clinical syndrome that is characterized by cognitive dysfunction in the elderly after surgery [1]. The incidence rate of POCD has reached up to 40% and largely increases mortality in the elderly [2]. POCD is also a risk factor for neurodegenerative disorders [3]. Although some drug candidates were tested in clinical trials, there are no effective drugs approved by the Food and Drug Administration for POCD [4]. It was previously reported that some signaling pathways related to neuronal apoptosis and neuroinflammation, such as the phosphatidylinositol-3-kinase (PI3-K)/protein kinase B (AKT), brain-derived neurotrophic factor, and inflammation-related pathways, were changed in both POCD patients and animals [5,6]. However, the detailed pathological mechanisms underlying POCD remain unclear, and there is still a lack of effective therapeutic targets for POCD [7]. Importantly, very few transcriptomic studies have focused on POCD. Only one study analyzed the expression profiles of mRNA in the serum of POCD and non-POCD patients [8], and the detailed relationship between these differentially expressed genes and the occurrence of POCD has remained uninvestigated or undisclosed. Therefore, it is urgent to systematically investigate the therapeutic targets of POCD.

Bioinformatics analysis could be used to establish biological models, clarify the biological processes of a certain disease, and screen for therapeutic targets [9]. A Connectivity Map (CMap) is a bioinformatics analytical tool that systematically matches similarities between pathological condition-related genes and perturbagens and is used to explore the connection among diseases, genes, and drugs [10]. Previously, a CMap was employed to screen therapeutic targets for neurological diseases, including ischemic stroke and Alzheimer’s disease (AD) [11].

In this study, a CMap model of POCD was established by using the differentially expressed genes of patients from previously reported transcriptomic data. Importantly, retinoic acid receptor (RAR), a transcriptional factor that has been shown to produce neuroprotective effects, was predicted to be a therapeutic target for POCD, and acitretin, a representative clinically used RAR agonist, was shown to effectively prevent surgery-induced cognitive impairments and induce the expression of RARα and RARα-target genes in aged mice.

## 2. Materials and Methods

### 2.1. Chemicals and Reagents

Fentanyl (fentanyl citrate injection, 0.05 mg/mL) was purchased from Humanwell Pharmaceutical Co., Ltd. (Yichang, China). Droperidol (droperidol injection, 2.5 mg/mL) was supplied by Sun Rise Pharmaceutical Co., Ltd. (Shanghai, China). Acitretin was obtained from Aladdin (Shanghai, China).

### 2.2. CMap Analysis of POCD Signature

The CMap database https://clue.io/ (accessed on 17 January 2022) collects various expression profiles of cultured human cells for the exploration of the relationship among genes, diseases, and drugs. Here, a CMap has been used to find the perturbation associated with the expression signatures of POCD. Source data regarding POCD-related small RNA transcriptomes were derived from the published literature [8]. Briefly, 200 patients requiring hip or knee replacement surgery were recruited. The neurological functions were measured at 1 day pre-surgery and 30 days post-surgery to identify post-surgery cognitive impairments in patients and RNA samples from serum of POCD and non-POCD patients were collected. Following this, 115 POCD-induced differential expressed mRNAs were evaluated by using microarray analysis. Among these differential expressed mRNAs, 14 up-regulated and 41 down-regulated genes were identified as POCD-related landmark genes for the establishment of a POCD signature of the CMap. The details of these landmark genes are described in Appendix A.

The CMap score, a CMap-specific parameter, was computed by the correlation between input signatures and every perturbation according to a previous study [12]. Briefly, Kolmogorov–Smirnov enrichment statistic was utilized to measure the similarity between the input signature and all the reference signatures caused by perturbagens in human cells. Then, the enrichment scores were normalized for meaningful comparisons of the connections across different cell types and perturbation types. In addition, the normalized enrichments were further standardized based on their distribution and eventually input as a CMap score. In our CMap analysis, the cell-summarized CMap score was summarized across all 9 cell lines for its higher efficiency and accuracy in searching novel targets.

The perturbagens, containing 2429, 3799, and 2160 signatures for compounds, gene knock-down (KD), and gene overexpression (OE), respectively, reflect the connection between the POCD signature and all perturbations. The CMap score of 8559 perturbagens ranges from −99.79 to 99.95, whereby a positive CMap score may indicate causes or exacerbations of the pathogenesis of POCD and a negative CMap score may indicate a therapeutic target against POCD.

### 2.3. Animals

Male ICR mice aged 12 months and weighing 30–40 g were obtained from Zhejiang Academy of Medical Sciences (Hangzhou, China). Mice were exposed to a 12 h light/dark cycle under humidity (50 ± 10%) and controlled temperature (22 ± 2 °C). Four to five animals were kept in each cage. Animals had free access to normal animal food (Shanghai Slac Laboratory Animal Co., Ltd., Shanghai, China) and water. All the procedures were performed according to the National Institutes of Health (NIH) Guide for the Care and Use of Laboratory Animals (NIH Publications No. 80-23, revised 1996) and were approved by the Animal Care and Use Committee of Ningbo University (SYXK-2019-0005).

Forty-two mice were allocated to four groups as follows: non-surgery + vehicle group, non-surgery + 10 mg/kg acitretin (dissolved in corn oil) group, surgery + vehicle group and surgery + 10 mg/kg acitretin group. Mice in non-surgery + vehicle and surgery + vehicle groups were treated with corn oil as the vehicle. Drugs or vehicle were administered by intraperitoneal (i.p.) injection once daily for 16 consecutive days post-surgery.

### 2.4. Anesthesia and Surgical Procedure

An established protocol for anesthesia and surgery was used with slight modification [13]. Mice were i.p. injected with a mixture of fentanyl (0.02 mg/kg) and droperidol (5.0 mg/kg). Then, the animals were returned to their cages and allowed to calm down in a familiar environment to facilitate the absorption the drugs. The adequacy of anesthesia was monitored by blink reflex, righting reflex, and paw withdrawal reflex.

After fentanyl and droperidol injection, mice were transferred to the surgery table. Briefly, the fur of the mice in the surgical site was shaved. A laparotomy was conducted using a 1.5-cm midline incision. Approximately 5 cm of the small intestine was removed from the peritoneal cavity, covered with a clean and moist gauze. After 3 min, the small intestine was replaced in the peritoneal cavity, and two layers of the abdominal wall were closed by sutures. Warming pads were used to maintain rectal temperature at around 37 °C throughout the procedure. All the procedures from the induction of anesthesia to the end of surgery lasted about 30 min.

### 2.5. Physiological Parameters Measurements

Six mice were allocated to two groups as follows: anesthesia group and anesthesia + surgery group. Then, the measurements of physiological parameters, including arterial O_2_ saturation, heart rate, breath rate, pulse distension, and rectal temperature were obtained by cervical collar telemetry (MouseOx, Starr Lifesciences, Oakmont, PA, USA). Physiological parameter measurements were applied after injection, and read at 0 (pre-operation), 10 (during-operation), and 20 min of anesthesia (post-operation), and 5 min after regaining consciousness (after-anesthesia).

### 2.6. Behavioral Tests

At day 6 post-surgery, open field tests were used to evaluate the locomotor and sensorimotor activities of animals [14]. The tests were conducted in a 50 × 50 × 39 cm open plastic box, the floor of which was divided into four equal quadrants by crossed black lines. Mice were placed in the center of the open field and allowed to explore freely for 5 min. The number of rearing (mice standing on their hind legs) and line crossing was recorded to analyze motor ability and the running speeds were recorded to reflect the sensorimotor functions of the mice. In order to avoid distribution of mice due to urine and odor, the open field was cleaned between two individual tests using 10% ethanol.

At day 7–9 post-surgery, novel object recognition (NOR) tests were conducted in an open-field arena (30 × 30 × 30 cm) constructed with polyvinyl chloride, plywood, and acrylic, exactly as described previously [13]. The task included acclimation, training, and retention over three consecutive days. On day 1, the animals were adapted to the experimental area for 5 min without exposure to any behavioral stimulus. On day 2, the animals explored two identical objects (black plastic cubes, 5 × 5 × 5 cm) for 5 min. On day 3, one of the objects was replaced by an object with a new shape and color (a gray plastic square pyramid, 5 × 5 × 7 cm) and the animals were again adapted to the area for 5 min. The field was decontaminated with 70% ethanol solution using a dry cloth between the tests. The animals explored the test area by sniffing or touching the objects with their nose and/or forepaws at a distance of less than 2 cm. Sitting or turning the objects was not considered exploratory behavior. Exploratory behavior was evaluated manually using a video camera by an observer blinded to the test conditions. Total exploration time refers to the amount of time devoted to the location of the two objects. Cognitive function was measured using a recognition index, which is the exploration time involving either of the two objects (training session) or the novel object (retention session) compared with the total exploration time.

At day 10 post-surgery, Y-maze tests were conducted in a 30 × 8 × 30 cm black open box with three identical arms according to a published protocol [15]. Different geometric figures were attached to particular arms as visual markers. The three arms were randomly assigned as the new arm and two other arms, and the junction was defined as the central area. The tests consisted of training and exploring sessions with an interval of 2 h. In the training session, the new arm was blocked by a partition. Mice were put into the central area and allowed to explore freely in the box, except the new arm, for 5 min. In the exploring session, the partition was removed. Mice were put into the central area and allowed to explore freely in the box for 5 min. The movement of mice was monitored by a video attached to a trajectory tracking system, and the time mice spent in each arm was recorded.

On days 11–16 post-surgery, Morris water maze tests were conducted to evaluate the spatial cognition of mice, as previously described [16]. The tests were carried out at room temperature in a 150-cm-diameter circle of water, which was divided into four equal quadrants. The circle platform was placed in the first quadrant, except for the last day. A computer-based video system was used to record the motion trails of mice in the water. The animals were trained to locate and reach the platform for five consecutive days. On the sixth day of the tests, the platform was removed as a probe trial and the mice were allowed to search the platform for 60 s. All data included the time mice spent to find the platform (five days). The speed, time, and entries mice made in the target quadrant, as well as the motion trails, were collected and analyzed.

### 2.7. Brain Tissue Harvest

On day 17 post-surgery, mice were deeply anesthetized with pentobarbital sodium (100 mg/kg, i.p.) and sacrificed by decapitation. Hippocampal regions were isolated quickly and stored at −80 °C for Western blotting analysis and quantitative real-time PCR (qRT-PCR), respectively.

### 2.8. Western Blotting Analysis

Western blotting analysis was conducted as previously described [16]. The protein was extracted from the hippocampal regions of mice with a lysis buffer at 4 °C for 1 min and centrifuged at 13,200 rpm for 30 min. The protein concentration in the supernatant was measured by Bradford assay. Samples were loaded on sodium dodecyl sulfate polyacrylamide gel electrophoresis and proteins were transferred to the polyvinylidene fluoride membranes. The membranes were blocked with 5% skim milk in tris buffered saline with Tween (TBST) for 2 h, followed by overnight incubation with primary antibodies against RARα (1:800, Cat No: A19551, Abclonal, Wuhan, China) and β-actin (1:5000, Cat No: AC026, Abclonal). The membranes were washed three times (15 min for each) with TBST and incubated with secondary antibodies HRP goat anti-rabbit IgG (Cat No: A0208, 1:3000, Beyotime, Shanghai, China) at room temperature for 45 min. Blots were visualized with enhanced chemiluminescence according to the manufacturer’s instructions (American Bioscience, Aylesbury, UK). Data were evaluated as optical density. Then, the expression of proteins was evaluated by Image J.

### 2.9. qRT-PCR

Total RNA was isolated from the hippocampal regions of mice using TRIzol reagent (Omega Bio-Tek, Norcross, GA, USA). qRT-PCR of RARα, phosphatase, and tensin homologue deleted on chromosome 10 (PTEN), a disintegrin and metalloprotease 10 (ADAM10), calbindin 1 (Calb1), and erb-b2 receptor tyrosine kinase 4 (Erbb4) were performed by using the Mx3005P qPCR System (Stratagene, La Jolla, CA, USA). β-actin mRNA was chosen as the internal control gene, and the relative expression of mRNA was analyzed through 2^−ΔΔCT^ method, which was used for calculations.

All primers were commercially available from BGI Tech Solutions (Beijing, China). The primer sequences and primer ID are listed in Table 1.

### 2.10. Data Analysis and Statistics

Data were expressed as means ± standard derivation (SD). One-way analysis of variance (ANOVA) was utilized to determine statistical significance, and post hoc multiple comparison was conducted through Tukey’s test. *p* < 0.05 was considered to have statistical significance.

## 3. Results

### 3.1. The Reliability and Predictability for the CMap Model of POCD

In the study, the reliability and predictability for the CMap model of POCD was investigated by evaluating the connection between POCD signature and known POCD-associated perturbagens. In 2429 compounds of the CMap, a significantly positive connectivity was identified in the majority of PI3-K inhibitors (Figure 1A) and AKT inhibitors (Figure 1B). Particularly, the CMap score of GDC-0941 (rank 14), AKT-inhibitor (rank 16), PI-828 (rank 18), and PIK-90 (rank 27) were 95.76, 95.29, 94.51 and 92.84, all at the top 30 of 2429 total compounds. Another 5959 perturbagens of OE and KD were analyzed in the CMap model of POCD. The CMap score of S100β OE was 56.07 (rank 451), and S100β KD was −66.9 (rank 5498, Figure 1C). Anti-POCD drug candidates listed on ClinicalTrials.gov were also screened in this CMap model of POCD. At the time, there were 246 drug intervention clinical trials registered to treat POCD. Among these, there were only 4 drugs employed, namely memantine, lidocaine, topiramate, and dexamethasone, matching the records in the CMap perturbagens. All these pre-clinical anti-POCD candidates displayed negative connections to POCD. For example, lidocaine had a −65.88 CMap score, ranking 2340 of the 2429 CMap compounds, and the −50.77 CMap score of memantines awarded it a ranking of 2240 in 2429 CMap compounds (Figure 1D).

### 3.2. RAR Was Predicated as a Novel Therapeutic Target for POCD

RAR is a nuclear receptor that actives the retinoic acid (RA) signaling pathway [17]. There is no prior study that has reported the association between RAR and POCD. Interestingly, strong negative correlations were found between RAR agonists and POCD (Figure 1E). In particular, the −94.38 CMap score of acitretin and −92.18 CMap score of AC-55649, two RAR agonists, ranked at 2422 and 2418 of the 2429 compounds in the CMap, respectively (Figure 1E). These results strongly suggested that RAR might be a novel therapeutic target for POCD.

### 3.3. Acitretin Alleviated Surgery-Induced Cognitive Dysfunction in Aged Mice

The physiological parameters of mice subjected to anesthesia and anesthesia + surgery groups were evaluated at various time points. The tested physiological parameters, including arterial O_2_ saturation, heart rate, breath rate and pulse distension, and rectal temperature were not significantly changed between mice with anesthesia and those with anesthesia + surgery at 0 (pre-operation), 10 (during-operation), and 20 min after anesthesia (post-operation), and 5 min after regaining consciousness (post-anesthesia), suggesting our protocol for establishing a POCD animal model did not induce the physiological changes in animals (Table 2).

The effects of acitretin, a representative RAR agonist, on cognition were further evaluated in this POCD animal model. The procedures of the behavioral tests are depicted schematically in Figure 2A. In the open field tests, no significant difference in the number of crossing and rearing among groups was observed, indicating no substantial alteration in motor function after surgery or acitretin treatment (one-way ANOVA, for crossing, F (3, 38) = 1.192, *p* = 0.3256, Figure 2B; for rearing, F (3, 38) = 0.02235, *p* = 0.9954, Figure 2C). In addition, the total running distance of mice was not altered among various groups in the open field tests (one-way ANOVA, F (3, 38) = 0.1193, *p* = 0.9482, Figure 2D), indicating that neither acitretin nor surgery induced sensorimotor deficits in animals. During the training phase of NOR tests, there was no detectable difference in recognition index among groups (one-way ANOVA, F (3, 38) = 2.111, *p* = 0.1149, Figure 2E), and in the retention session, the recognition index was statistically different among groups (one-way ANOVA, F (3, 38) = 9.242, *p* = 0.0001, Figure 2F). Surgery significantly decreased the recognition index in the retention session (Tukey’s test, *p* = 0.0005, Figure 2F). Interestingly, a significant increase in recognition index was evidenced between surgery + vehicle and surgery + acitretin groups (Tukey’s test, *p* = 0.0003, Figure 2F), while no significant difference was found between the non-surgery + vehicle and non-surgery + acitretin groups (Tukey’s test, *p* = 0.9830, Figure 2F). These results suggested that acitretin attenuated surgery-induced spatial cognitive dysfunction in aged mice. In Y-maze tests, mice in the surgery + vehicle group significantly decreased the entries in the novel arms compared to those in non-surgery + vehicle group (one-way ANOVA, F (3, 38) = 3.691, *p* = 0.02; Tukey’s test, *p* = 0.0455, Figure 2G), and acitretin significantly attenuated a surgery-induced decrease in the entries in the novel arms in the Y-maze tests (Tukey’s test, *p* = 0.0297, Figure 2G). On the fifth day of the training session in the Morris water maze tests, compared with the non-surgery + vehicle group, the time of reaching the platform was significantly prolonged in the surgery + vehicle group (one-way ANOVA, F (3, 38) = 6.063, *p* = 0.0018, Tukey’s test, *p* = 0.009, Figure 2K,L). Moreover, the escape latency showed no difference between the non-surgery + vehicle and non-surgery + acitretin groups (Tukey’s test, *p* = 0.9988, Figure 2K,L), and the escape latency was significantly decreased in the surgery + acitretin group compared to the surgery + vehicle group, indicating that acitretin ameliorated surgery induced spatial memory impairments in aged mice (Tukey’s test, *p* = 0.0033, Figure 2K,L). The swimming speed of mice was not significantly changed among groups in the probe trial of the Morris water maze tests (one-way ANOVA, F (3, 38) = 0.7621, *p* = 0.5224, Figure 2H). Moreover, the time and entries in the target quadrant were significantly different among groups (one-way ANOVA, for percentage of time in the target quadrant, F (3, 38) = 4.647, *p* = 0.0073, Figure 2I; for percentage of entries in the target quadrant, F (3, 38) = 6.214, *p* = 0.0015, Figure 2J). A significant decrease in the percentage of time and entries in the target quadrant was observed in the surgery + vehicle group compared to the non-surgery + vehicle group (Tukey’s test, for percentage of time in the target quadrant, *p* = 0.0371, Figure 2I; Tukey’s test, for percentage of entries in the target quadrant, *p* = 0.0028, Figure 2J). Moreover, the percentage of time and entries in the target quadrant was higher in the surgery + acitretin group than those in the surgery + vehicle group (Tukey’s test, for percentage of time in the target quadrant, *p* = 0.0189, Figure 2I; Tukey’s test, for percentage of entries in the target quadrant, *p* = 0.0143, Figure 2J). These results suggested that acitretin alleviated surgery-induced cognitive dysfunction in aged mice.

### 3.4. Acitretin Attenuated the Reduction of RARα Expression and Increased mRNA Expression of RARα-Target Genes in Aged Mice after Surgery

The level of RARα protein was significantly decreased in the surgery + vehicle group, and acitretin significantly prevented the surgery-induced reduction in RARα expression in the hippocampal regions of aged mice (one-way ANOVA F (3, 8) = 38.36, *p* < 0.0001, Tukey’s test, *p* = 0.0334, Figure 3A,B). In addition, the mRNA expression of RARα was elevated by acitretin in aged mice after surgery (one-way ANOVA, F (3, 8) = 34.65, *p* < 0.0001, Tukey’s test, *p* = 0.035, Figure 3C). Compared with the non-surgery + vehicle group, the mRNA expression of RARα-target genes, namely PTEN, ADAM10, Calb1, and Erbb4, were significantly decreased in the surgery + vehicle group (Tukey’s test, for PTEN, *p* = 0.0061, Figure 4A; for ADAM10, *p* = 0.0149, Figure 4B; for Calb1, *p* = 0.0030, Figure 4C; for Erbb4, *p* = 0.0065, Figure 4D). Moreover, acitretin significantly prevented the surgery-induced reduction in mRNA expression of RARα-target genes (Tukey’s test, for PTEN, *p* = 0.0142, Figure 4A; for ADAM10, *p* = 0.0319, Figure 4B; for Calb1, *p* = 0.0006, Figure 4C; for Erbb4, *p* = 0.0468, Figure 4D). These results suggested that acitretin produced anti-POCD effects by activating RARα and inducing RARα-target genes in the hippocampal regions of animals.

## 4. Discussion

In this study, a reliable CMap model of POCD was established. In addition, RAR was predicted to be a target for POCD by this CMap model, which was validated by the evidence that acitretin, a representative RAR agonist, prevented surgery-induced cognitive dysfunction and reduction in RARα signaling in aged mice (Figure 5).

POCD, a postoperative syndrome of cognitive impairments with high prevalence, remains a major clinical problem because of its unclear pathogenesis and the global lack of effective therapies. The CMap is based on massive transcriptional signatures of perturbagens, and it can generate substantial information by utilizing only the changes in 978 landmark genes [12]. Therefore, the CMap analysis is appropriate for studying underlying pathological mechanisms of diseases with relatively little established knowledge. Considering that transcriptional signatures in the CMap are derived from human cell lines, it was decided to screen POCD patient-derived transcriptomic data previously collected and available in PubMed. The only human POCD study in mRNA transcriptome recruited aged patients diagnosed with POCD, from whom serum samples were collected for evaluating microarray expression profiles [8]. The differentially expressed mRNA was selected from that study, and 55 of these genes are landmark genes in the CMap. Previous studies have shown that a predictable and reliable CMap model of disease could be established by using 50–70 differently expressed landmark genes. For example, a CMap model of hypoxia in neuroblastoma has been established previously using 52 landmark genes [18]. A different CMap model was produced for thyroid-associated orbitopathy using 54 landmark genes, and this predicted 6 novel candidates that were further validated in vitro [19]. Therefore, we have established a CMap model of POCD by using 55 differently expressed landmark genes.

In our study, the positive CMap scores of PI3-K and AKT inhibitors indicated that the inhibition of the PI3-K/AKT pathway might be one of the major inducers of POCD, and these results are consistent to many previous studies [20,21]. In addition, the positive connection between S100β OE and POCD and the negative connection between S100β KD and POCD suggested that S100β might be another inducer of POCD, which was supported by previous reports [22,23]. These results also serve as a validation for the predictability of our model. Furthermore, currently-used anti-POCD drugs were negatively correlated with POCD, implying our CMap model of POCD is predictable, and might be useful for drug repurposing or reposition.

In the present study, RAR was predicted to be a potential target for POCD because RAR agonists were strongly and negatively correlated with POCD. To the best of our knowledge, there is no relationship between POCD and RAR established in the literature. RAR has extensively characterized physiological effects. After binding with RA, RAR acts on the retinoic acid response elements in the nucleus, promoting the translation of RAR-responsive genes [24]. RAR plays a crucial role in cognitive function by promoting neuronal growth, survival, and synaptic plasticity [24]. Vitamin A deficiency inactivated RAR, and thus induced cognitive impairments [25]. Moreover, the downregulation of RA signaling is observed during aging and in neurodegeneration, indicating that the activation of RAR might be used to treat neurological diseases [24,26]. RAR agonists decrease β-amyloid (Aβ) toxicity in AD mice [27]. Patients with POCD have a high risk for developing neurodegenerative disorders, indicating that POCD possibly shares similar pathological pathways to neurodegenerative disorders. The anti-POCD effects of acitretin, a representative RAR agonist that is negatively correlated with POCD, was tested to validate RAR as a potential therapeutic target in POCD. The general safety profile of acitretin has been demonstrated in its clinical application for psoriasis. Moreover, acitretin readily cross the blood–brain barrier, indicating that this drug might be used as a neurological agent [28]. In a Phase II randomized clinical trial in Germany, oral administration of acitretin significantly increased APPs-α levels in the cerebrospinal fluid of AD patients, indicating the potential use of this drug to treat AD [29]. In the present study, 10 mg/kg acitretin was employed because it was earlier reported to safely treat AD animals. Our results revealed that acitretin significantly prevented surgery-induced cognitive impairments in aged mice, providing a support that RAR is a potential target for POCD.

RAR has three subtypes, namely RARα, RARβ, and RARγ. RARα, but not RARβ or RARγ, was mainly expressed in the hippocampal region, a critical brain region determining cognitive and memory functions [30]. Therefore, we speculated that acitretin might produce anti-POCD cognitive-enhancing effects via the prevention of surgery-induced reduction in RARα, and the induction of RARα-target genes in the hippocampal regions of animals. ADAM10, PTEN, Calb1, and Erbb4 were reported as RARα-target genes to regulate cognition. The activation of ADAM10 could prevent Aβ-induced neurotoxicity in AD animals [31]. The induction of PTEN promoted synaptogenesis and synaptic plasticity in brains, leading to the enhancement of cognition [32]. The activation of Calb1, a calcium-binding protein widely expressed in neurons, increased neuronal viability and reduced neuronal apoptosis in vascular dementia rats [33]. Moreover, the decreased expression of Erbb4 inhibited nerve regeneration and participated in peripheral nerve injuries postoperatively [34]. In the present study, the expression of these RARα-target genes was downregulated by surgery, and reversed by acitretin. In addition, acitretin prevented the surgery-induced reduction in RARα in the hippocampal regions of aged mice. These results suggested that acitretin might prevent POCD-induced cognitive impairments by activating RARα and RARα-target genes.

Previous studies have reported that many vitamin A metabolites could reduce oxidative stress via increasing the transcription of downstream anti-oxidant genes [35,36]. Oxidative stress further induces neurodegeneration during POCD, and might be inhibited by antioxidants [37]. Although the detailed mechanisms underlying anti-POCD effects of vitamin A metabolites have not be investigated in this study, we speculated that one of the mechanisms of action for vitamin A metabolites to antagonize POCD is to decrease oxidative damage by acting on RAR. This speculation is based on the evidence that PTEN, an RAR-target gene, could produce anti-oxidative stress effects via maintaining the activity of cyclooxygenase [38], and the activation of Erbb4, another RAR-target gene, was demonstrated to reduce the generation of ROS in the brain [39].

Given that POCD is a complicated problem, it is not likely that RAR is a single pathway that is solely responsible for the underlying pathology of POCD. For example, the inverse relationship between vitamin D and POCD is well known, and retinoid X receptor (RXR) served as a partner for both vitamin D receptor (VDR) and RAR [9,10,11]. It is possible that the overactivated RAR might competitively bind to RXR, leading to the antagonization of the anti-POCD effects of vitamin D [12,13,14]. Similarly, the overactivation of VDR might also inhibit the function of RAR via disrupting RAR-RXR heterodimerization. Excessive vitamin A and vitamin D were reported to produce neurotoxicity [15,16]. High doses of vitamin A and retinoids could increase the occurrence of neuroinflammation, oxidative stress, and mitochondrial dysfunction in the brain [17,18,19]. Moreover, excess intake of vitamin D might induce neuropsychiatric symptoms, such as attention deficit, apathy, confusion, and drowsiness, in the aged population [15,20,21]. Therefore, it is reasonable to prevent POCD by supplying appropriate amounts of RAR and VDR agonists, maintaining the balanced activities of RAR-RXR and VDR-RXR.

## 5. Conclusions

In conclusion, to the best of our knowledge, this study is the first to reveal the connection between RAR and POCD. The study also denotes a reliable CMap bioinformatics model of POCD for future use and establishes that RAR is a novel therapeutic target for treating this clinical syndrome.

## Figures and Tables

**Figure 1 pharmaceutics-15-02311-f001:**
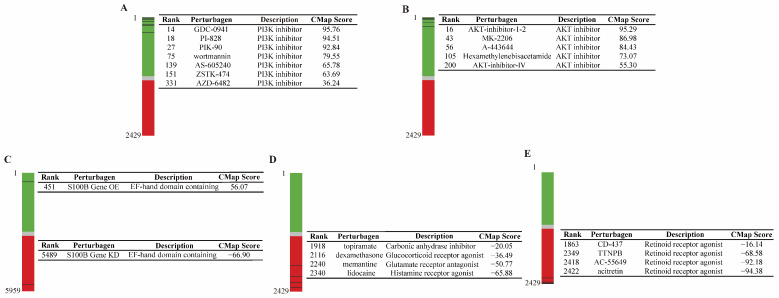
CMap model of POCD reveals the association between perturbagens and POCD. (**A**) Positive connections between PI3-K inhibitors and POCD was found in our CMap model of POCD. (**B**) Positive connections between AKT inhibitors and POCD was found in our CMap model of POCD. (**C**) A positive connection between OE of S100β and POCD, and a negative connection between KD of S100β and POCD was revealed in our CMap model of POCD. (**D**) Negative connections between four anti-POCD candidates and POCD were found in our CMap model of POCD. (**E**) Negative connections between RAR agonists and POCD were discovered in our CMap model of POCD.

**Figure 2 pharmaceutics-15-02311-f002:**
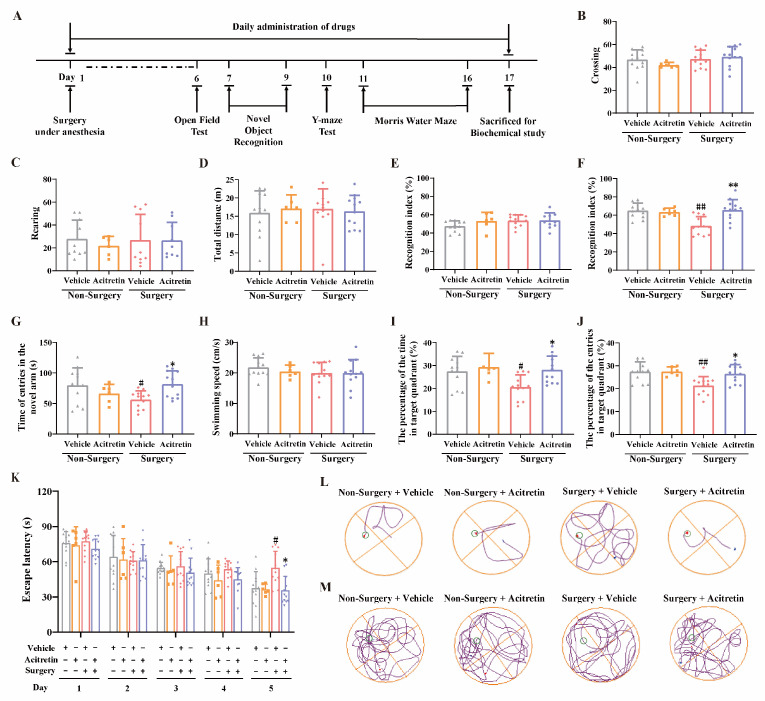
Acitretin ameliorates cognitive impairments induced by abdominal surgery under anesthesia in aged mice. (**A**) Chronological course of the experimental design. (**B**–**D**) In open field tests, acitretin did not affect the number of line crossing (**B**), rearing (**C**), and total running distance (**D**) of aged mice. (**E**,**F**) In the training session of NOR tests, acitretin did not affect the recognition index (**E**), while in retention session, acitretin significantly attenuated the reduction of recognition index in aged mice after surgery (**F**). (**G**) In Y-maze tests, acitretin prevented surgery-induced decrease in entries in the novel arms in aged mice. (**H**–**J**) During the probe trial of Morris water maze tests, the swimming speeds among groups were not significantly altered (**H**). Acitretin significantly mitigated the decrease in the percentage of time in the target quadrant (**I**), and percentage of entries in the target quadrant (**J**) in the aged mice after surgery. (**K**) In the fifth day of the training periods of Morris water maze tests, acitretin significantly reduced the escape latency induced by the surgery. (**L**,**M**) Representative swimming paths of mice in various groups at the last day in the training periods of Morris water maze tests were shown in (**L**), while the representative paths of mice in various groups in the probe trial of Morris water maze tests were shown in (**M**). The data were expressed with mean ± SD (*n* = 6–12 for each group), ^#^ *p* < 0.05 and ^##^ *p* < 0.01 vs. non-surgery + vehicle group, * *p* < 0.05 and ** *p* < 0.01 vs. surgery + vehicle group (one-way ANOVA and Tukey’s test).

**Figure 3 pharmaceutics-15-02311-f003:**
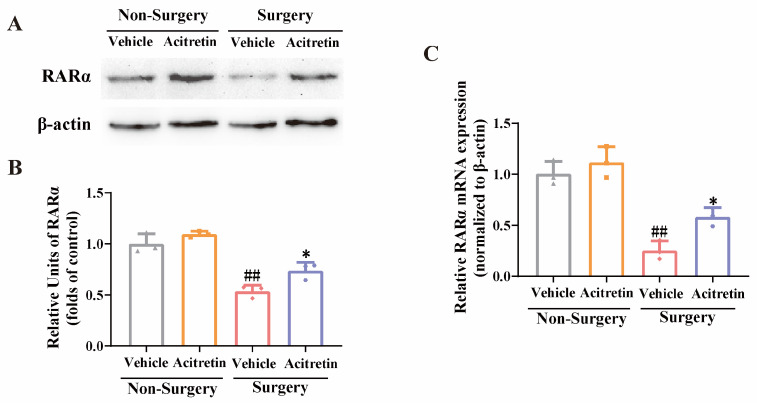
Acitretin inhibited surgery-induced decrease in RARα expression at protein and mRNA levels in the hippocampal regions of aged mice. (**A**) The representative blots of RARα expressed in the hippocampal regions of aged mice. (**B**) Quantitative results showed that acitretin significantly alleviated surgery-induced decrease in the protein level of RARα in the hippocampal regions of aged mice. (**C**) qRT-PCR showed that acitretin significantly attenuated surgery-induced down-regulation of RARα mRNA in the hippocampal regions of the aged mice. The data were expressed with mean ± SD (*n* = 3 for each group), ^##^ *p* < 0.01 vs. non-surgery + vehicle group, and * *p* < 0.05 vs. surgery + vehicle group (one-way ANOVA and Tukey’s test).

**Figure 4 pharmaceutics-15-02311-f004:**
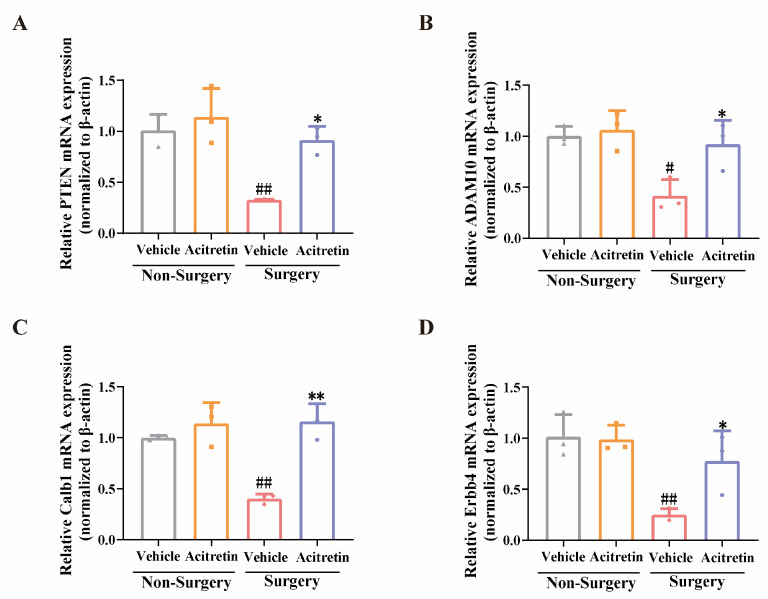
Down-regulation of RARα-target mRNA expression induced by surgery was reserved by acitretin in the hippocampal regions of aged mice. The expression of (**A**) PTEN, (**B**) ADAM10, (**C**) Calb1 and (**D**) Erbb4 were measured by qPCR in the hippocampal regions of aged mice. The data were expressed with mean ± SD (*n* = 3 for each group), ^#^ *p* < 0.05 and ^##^ *p* < 0.01 vs. non-surgery + vehicle group, and * *p* < 0.05 and ** *p* < 0.01 vs. surgery + vehicle group (one-way ANOVA and Tukey’s test).

**Figure 5 pharmaceutics-15-02311-f005:**
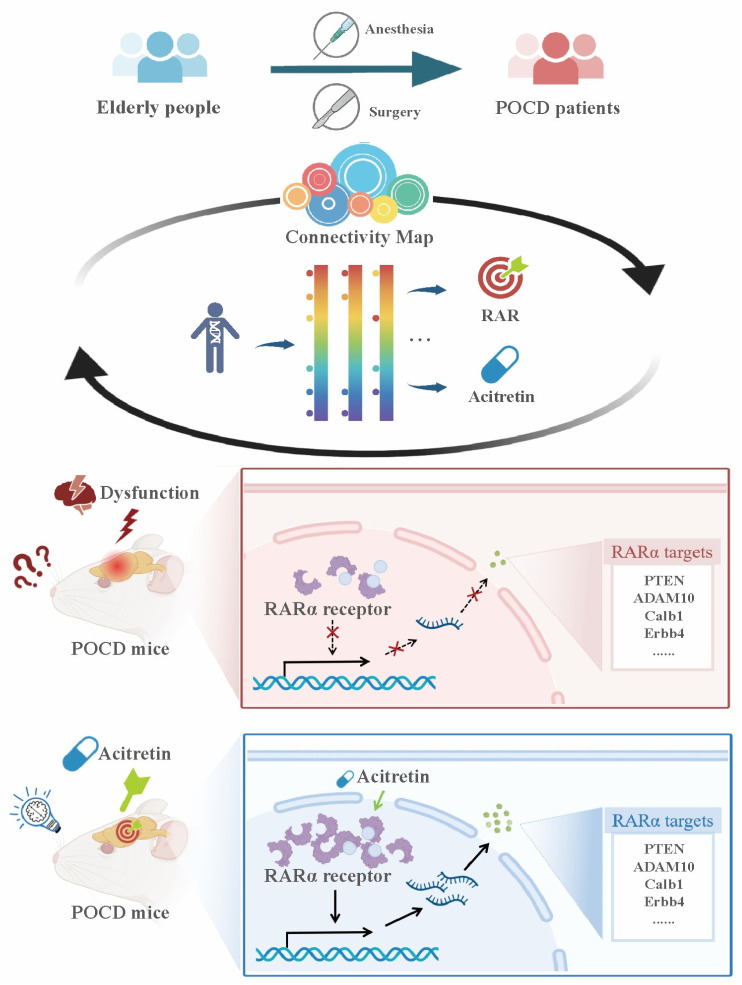
A reliable CMap model of POCD was established. RAR was predicted to be a target for POCD by this CMap model, which was validated by the evidence that acitretin, a representative RAR agonist, prevented surgery-induced cognitive dysfunction and reduction of RARα signaling in aged mice.

**Table 1 pharmaceutics-15-02311-t001:** Primers sequences for qRT-PCR.

Genes	Primer ID	Sequences
RARα	220406_001A01	Forward: 5′-TTCTTTCCCCCTATGCTGGGT-3′
	220406_001B01	Reverse: 5′-GGGAGGGCTGGGTACTATCTC-3′
PTEN	211018_005C02	Forward: 5′-TGGATTCGACTTAGACTTGACCT-3′
	211018_005D02	Reverse: 5′-GCGGTGTCATAATGTCTCTCAG-3′
ADAM10	211018_005E01	Forward: 5′-GGGAAGAAATGCAAGCTGAA-3′
	211018_005F01	Reverse: 5′-CTGTACAGCAGGGTCCTTGAC-3′
Calb1	211018_005A02	Forward: 5′-TCTGGCTTCATTTCGACGCTG-3′
	211018_005B02	Reverse: 5′-ACAAAGGATTTCATTTCCGGTGA-3′
Erbb4	211018_005G01	Forward: 5′-CCTTCCTGCGGTCTATCCGA-3′
	211018_005H01	Reverse: 5′-CCAAAGTTGCCATCTTTCCTGTA-3′
β-actin	220725_002B07	Forward: 5′-CCTAAGAGGAGGATGGTCGCR-3′
	220725_002C07	Reverse: 5′-CCTAAGAGGAGGATGGTCGC-3′

**Table 2 pharmaceutics-15-02311-t002:** Physiological parameters of mice subjected to anesthesia and anesthesia + surgery.

Parameters	Group	Anesthesia Duration (min)			Post-Anesthesia
		0	10	20	
Arterial O_2_ saturation (%)	Anesthesia	94.70 ± 1.15	94.64 ± 3.80	94.94 ± 3.50	96.17 ± 2.44
	Anesthesia + Surgery	92.90 ± 0.40	93.20 ± 1.02	95.85 ± 2.23	94.46 ± 1.19
Heart rate (bpm)	Anesthesia	413.6 ± 52.8	443.5 ± 82.8	539.1 ± 117.3	549.1 ± 136.9
	Anesthesia + Surgery	541.3 ± 103.1	532.8 ± 80.5	527.5 ± 60.1	511.3 ±70.2
Breath rate (brpm)	Anesthesia	133.4 ± 35.1	147.5 ± 45.2	112.7 ± 7.74	124.0 ± 9.29
	Anesthesia + Surgery	112.2 ± 11.3	112.7 ± 18.3	107.2 ± 17.2	139.8 ± 67.9
Pulse distention (μm)	Anesthesia	517.3 ± 150.3	463.2 ± 152.2	487.3 ± 192.5	396.9 ± 157.7
	Anesthesia + Surgery	431.5 ± 193.1	425.6 ± 182.7	435.2 ± 237.8	397.3 ± 243.5
Rectal temperature (°C)	Anesthesia	38.45 ± 0.86	38.18 ± 0.77	38.37 ± 0.20	38.99 ± 0.20
	Anesthesia + Surgery	38.17 ± 0.33	38.56 ± 0.39	38.60 ± 0.44	38.72 ± 0.48

Note: Values are mean ± SD (*n* = 3 for each group). Surgery was performed at 5–15 min post anesthesia. Physiological parameters were monitored at 0 (pre-operation), 10 (during-operation), and 20 min after anesthesia (post-operation), and 5 min after regaining consciousness (post-anesthesia). bpm: beats per min; brpm: breaths per min.

## Data Availability

Not applicable.

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
