# Peer review of "Retinoic Acid Receptor Is a Novel Therapeutic Target for Postoperative Cognitive Dysfunction"

_pharmaceutics, 2023, doi:10.3390/pharmaceutics15092311_

Round 1

Reviewer 1 Report

The study is well done and well-structured and presented. However, it could be improved by including two ideas or concepts, even if only in the discussion:

i) The antioxidant properties of various vitamin A metabolites are well known. Since postoperative cognitive dysfunction (POCD) has been linked to the presence of oxidative stress, it might be interesting to suggest that one of the mechanisms of action to antagonise POCD is to decrease oxidative damage by acting on the retinoic acid receptor (RAR).

ii) The inverse relationship between vitamin D and POCD is well known (there is even a meta-analysis of the relationship between vitamin D and POCD [Kuo-Chuan Hung et al, J Clin Anesth 2022]). However, the nuclear transcription factor (RXR) is a partner of both vitamin D receptor (VDR) and retinoic acid receptor (RAR). VDR ligand of 1,25(OH)2D and retinoic acid, respectively, employ the same heterodimerization partner (RXR) for binding to response elements of target genes. Thereby, it is possible that high concentrations of either retinoic acid or 1,25(OH)2D might reduce the availability of RXR and may antagonize the effects on each other compound. Reported evidence in animal models and clinical data in humans support this concept. Therefore, to what extent might it be desirable to act on the RAR to antagonise postoperative cognitive dysfunction (POCD), if we could be attenuating the expression of VDR?

Author Response

Response to Reviewer 1 Comments

We appreciate the time and efforts from your reviewers to help us to improve our manuscript. We have upleaded the revised manuscript as the attachment file. In our revised manuscript, we have dealt with all comments in a point-by-point fashion per your instructions.

Comments 1: The antioxidant properties of various vitamin A metabolites are well known. Since postoperative cognitive dysfunction (POCD) has been linked to the presence of oxidative stress, it might be interesting to suggest that one of the mechanisms of action to antagonize POCD is to decrease oxidative damage by acting on the retinoic acid receptor (RAR).

Response 1: Thank you for your valuable suggestion. Previous studies have reported that many vitamin A metabolites could reduce oxidative stress via increasing the transcription of downstream anti-oxidant genes [1,2]. Oxidative stress could further induce neurodegeneration during POCD, and might be inhibited by antioxidants [3]. Although the detailed mechanisms underlying the anti-POCD effects of vitamin A metabolites have not be investigated in this study, as suggested by the reviewer, we speculated that one of the mechanisms of action for vitamin A metabolites to antagonize POCD is to decrease oxidative damage by acting on retinoic acid receptor (RAR). This speculation is also based on the evidence that PTEN, an RAR-target gene, could produce anti-oxidative stress effects via maintaining cyclooxygenase activity [4]. And the activation of Erbb4, another RAR-target gene, was demonstrated to reduce the generation of ROS in the brain [5]. We have accordingly added these descriptions in our revised manuscript.

Comments 2: The inverse relationship between vitamin D and POCD is well known (there is even a meta-analysis of the relationship between vitamin D and POCD [Kuo-Chuan Hung et al, J Clin Anesth 2022]). However, the nuclear transcription factor (RXR) is a partner of both vitamin D receptor (VDR) and retinoic acid receptor (RAR). VDR ligand of 1,25(OH)2D and retinoic acid, respectively, employ the same heterodimerization partner (RXR) for binding to response elements of target genes. Thereby, it is possible that high concentrations of either retinoic acid or 1,25(OH)2D might reduce the availability of RXR and may antagonize the effects on each other compound. Reported evidence in animal models and clinical data in humans support this concept. Therefore, to what extent might it be desirable to act on the RAR to antagonise postoperative cognitive dysfunction (POCD), if we could be attenuating the expression of VDR?

Response 2: It is a critical comment. As pointed out by the reviewer, the inverse relationship between vitamin D and POCD is well known, and retinoid X receptor (RXR) is a partner of both vitamin D receptor (VDR) and RAR [6,7]. It is possible that the overactivated RAR might competitively bind to RXR, leading to the antagonization of the anti-POCD effects of vitamin D [8]. Similarly, the overactivated VDR might also inhibit the function of RAR via disrupting RAR-RXR heterodimerization. Actually, excessive vitamin A and vitamin D were reported to produce neurotoxicity. High doses of vitamin A and retinoids could increase the occurrence of neuroinflammation, oxidative stress and mitochondrial dysfunction in brain [9,10]. Moreover, excess intake of vitamin D might induce neuropsychiatric symptoms, such as attention deficit, apathy, confusion, drowsiness, in aged population [11,12]. Therefore, it is reasonable to prevent POCD by supplying appropriate amounts of RAR and VDR agonists, keeping the balanced activity of RAR-RXR and VDR-RXR. We have accordingly added these descriptions in our revised manuscript.

Reference:

  1. Sierra-Mondragon, E.; Rodríguez-Muñoz, R.; Namorado-Tonix, C.; Molina-Jijon, E.; Romero-Trejo, D.; Pedraza-Chaverri, J.; Reyes, J.L. All-Trans Retinoic Acid Attenuates Fibrotic Processes by Downregulating TGF-β1/Smad3 in Early Diabetic Nephropathy. Biomolecules 2019, 9.
  2. Singh, A.B.; Guleria, R.S.; Nizamutdinova, I.T.; Baker, K.M.; Pan, J. High glucose-induced repression of RAR/RXR in cardiomyocytes is mediated through oxidative stress/JNK signaling. J Cell Physiol 2012, 227, 2632-2644.
  3. Lin, X.; Chen, Y.; Zhang, P.; Chen, G.; Zhou, Y.; Yu, X. The potential mechanism of postoperative cognitive dysfunction in older people. Exp Gerontol 2020, 130, 110791.
  4. Wang, P.; Li, R.; Yuan, Y.; Zhu, M.; Liu, Y.; Jin, Y.; Yin, Y. PTENα is responsible for protection of brain against oxidative stress during aging. Faseb j 2021, 35, e21943.
  5. Xu, J.; Hu, C.; Chen, S.; Shen, H.; Jiang, Q.; Huang, P.; Zhao, W. Neuregulin-1 protects mouse cerebellum against oxidative stress and neuroinflammation. Brain Res 2017, 1670, 32-43.
  6. Hung, K.C.; Wang, L.K.; Lin, Y.T.; Yu, C.H.; Chang, C.Y.; Sun, C.K.; Chen, J.Y. Association of preoperative vitamin D deficiency with the risk of postoperative delirium and cognitive dysfunction: A meta-analysis. J Clin Anesth 2022, 79, 110681.
  7. Zhang, J.; Zhang, X.; Yang, Y.; Zhao, J.; Yu, Y. Correlation Analysis of Serum Vitamin D Levels and Postoperative Cognitive Disorder in Elderly Patients With Gastrointestinal Tumor. Front Psychiatry 2022, 13, 893309.
  8. Huang, P.; Chandra, V.; Rastinejad, F. Retinoic acid actions through mammalian nuclear receptors. Chem Rev 2014, 114, 233-254.
  9. de Oliveira, M.R.; da Rocha, R.F.; Pasquali, M.A.; Moreira, J.C. The effects of vitamin A supplementation for 3 months on adult rat nigrostriatal axis: increased monoamine oxidase enzyme activity, mitochondrial redox dysfunction, increased β-amyloid(1-40) peptide and TNF-α contents, and susceptibility of mitochondria to an in vitro H2O2 challenge. Brain Res Bull 2012, 87, 432-444.
  10. Hathcock, J.N.; Hattan, D.G.; Jenkins, M.Y.; McDonald, J.T.; Sundaresan, P.R.; Wilkening, V.L. Evaluation of vitamin A toxicity. Am J Clin Nutr 1990, 52, 183-202.
  11. Cui, X.; Eyles, D.W. Vitamin D and the Central Nervous System: Causative and Preventative Mechanisms in Brain Disorders. Nutrients 2022, 14.
  12. Lima, G.O.; Menezes da Silva, A.L.; Azevedo, J.E.C.; Nascimento, C.P.; Vieira, L.R.; Hamoy, A.O.; Oliveira Ferreira, L.; Bahia, V.; Muto, N.A.; Lopes, D.C.F.; et al. 100 YEARS OF VITAMIN D: Supraphysiological doses of vitamin D changes brainwave activity patterns in rats. Endocr Connect 2022, 11.

Reviewer 2 Report

The authors investigated the role of retinoic acid receptor alpha (RARα) in postoperative cognitive dysfunction (POCD) that occurs in elders subjected to the surgery, as well as the effects of potential treatment with acitretin since the effective treatment for this clinical syndrome is currently unavailable. For the purpose of the study they established connectivity Map (CMap) model and also used ICR mice to test behavior (open field test, novel object recognition test (NOR), Y-maze test, Morris water maze test), levels of physiological parameters (arterial O2 saturation, heart rate, breath rate, pulse distension and rectal temperature) and the expression of RARα using Western blotting analysis and RARα, phosphatase and tensin homologue deleted on chromosome 10 (PTEN), a disintegrin and metalloprotease 10 (ADAM10), Calbindin 1 (Calb1) and Erbb4 by qRT-PCR. The findings indicate that CMap bioinformatics model is reliable for POCD while RAR is a novel therapeutic target for treating POCD. Although the study is of sufficient significance and originality, there are numerous issues that need to be addressed:

1. Section Abstract should be rewritten since it is poorly fitted and confusing. For instance, what were the experimental groups? It is, also, unclear which types of samples were analyzed by which method. Some sentences are too general, like the sentence: “Connectivity Map (CMap) is a bioinformatics tool that aids in the creation and analysis of datasets after perturbation by drugs and diseases.” It should be shorten and compressed with the next sentence. These are just some examples of the shortcomings of this section. It is necessary to read it thoroughly and rewrite it to be much more fluent and informative.

2. In section Material and Methods the descriptions of the methods should be arranged in a logical order, those specific for the serum samples of POCD and non-POCD patients then those specific for the animals. It is necessary to better define the experimental groups. What is the control group? Are those animals treated with vehicle or they were subjected to the sham operation and treated with vehicle? The information about the time and the way of sacrifice of the animals is missing in the text (it is only provided on the Figure 2A). Which tissue was used for the Western blot analysis? Which secondary antibodies were used? What were the catalog numbers of the used primers as well as they producer? When were the behavioral tests performed? This information should be specified in the section Materials and methods, subsection Behavioral tests for each test. The authors should perform additional experiments for assessment of sensorimotor deficit, and novel object location test, as well as other time points for NOR test, to confirm presented results. At the beginning of the subsection 2.3. CMap analysis of POCD signature the detailed explanation of the method is provided while that information is missing for other listed methods. When were the behavioral tests performed? This information should be specified in the section Materials and methods, subsection Behavioral tests for each test. What correlation test was used? The reference for NOR test is not provided where it is indicated, but at it was introduced at the end of the paragraph. In this section, it should be emphasized that the schematic representation of the study is shown in Figure 2A. These are just some examples of the shortcomings of this section. It is necessary to read it thoroughly and rewrite and arranged it to be much more fluent and informative.

3. In section Results the positive and negative correlations are mentioned, but without the correlation coefficient. The information should be provided. The explanation why each method was performed was provided in this section, it is better to transfer those sentences into sections Materials and Methods, or even Discussion.

4. Section Discussion is not thorough enough and the key points are not well discussed. The section is written too general without logical connection with the obtained results of this study. For instance, the part of the paragraph is dedicated to PI3-K/AKT signaling, and then the other part is about the drugs for POCD treatment in general. It is unclear whether PI3-K/AKT signaling is altered after the drugs and why is that important for the current study. A very short paragraph about the possible connection between RAR and PI3-K/Akt pathway is provided only at the end of this section, without detailed explanation. Moreover, is there association between RAR and listed treatments, like dexamethasone, memantine, etc?

The references are missing, including for the sentence “It was previously reported that the activation of RAR could lead to the activation of the PI3-K/Akt pathway in neurons, indicating a crosstalk between RAR and the PI3-K/Akt pathway in the pathogenesis of POCD.”

Several sentences, including the sentence: “Consistent with these earlier reports, in the present study that the inhibitors of the PI3-K/AKT pathway were positively correlated with POCD and that the inhibition of this pathway might contribute to the pathogenesis of POCD.” are unclear and confusing. Overall, this section should be improved and more concisely written so the findings and their interpretations are better linked to the results of prior published studies, while the obtained/presented results should be discussed in better manner.

5. In the Legend of the Figure 3A the representative WB are presented, not what is indicated below the Figure.

6. The abbreviations should be defined when first used in the text, and once introduced the abbreviations should be used through the manuscript in the same manner (for instance NOR, CSF, etc). The authors should read the manuscript thoroughly and uniform this item.

7. It is necessary to pay attention to lower and upper case letters, as well as punctuation marks.

Minor editing of English language required

Author Response

Response to Reviewer 2 Comments

We appreciate the time and efforts from your reviewers to help us to improve our manuscript. We have dealt with all comments in a point-by-point fashion per your instructions and submitted the revised manuscript as an attachment.

Comments 1: Section Abstract should be rewritten since it is poorly fitted and confusing. For instance, what were the experimental groups? It is, also, unclear which types of samples were analyzed by which method. Some sentences are too general, like the sentence: “Connectivity Map (CMap) is a bioinformatics tool that aids in the creation and analysis of datasets after perturbation by drugs and diseases.” It should be shorten and compressed with the next sentence. These are just some examples of the shortcomings of this section. It is necessary to read it thoroughly and rewrite it to be much more fluent and informative.

Response 1: Thank you for your suggestion. We have accordingly rewritten Abstract in the revised manuscript.

Comments 2: In section Material and Methods the descriptions of the methods should be arranged in a logical order, those specific for the serum samples of POCD and non-POCD patients then those specific for the animals. It is necessary to better define the experimental groups. What is the control group? Are those animals treated with vehicle or they were subjected to the sham operation and treated with vehicle? The information about the time and the way of sacrifice of the animals is missing in the text (it is only provided on the Figure 2A). Which tissue was used for the Western blot analysis? Which secondary antibodies were used? What were the catalog numbers of the used primers as well as they producer? When were the behavioral tests performed? This information should be specified in the section Materials and methods, subsection Behavioral tests for each test. The authors should perform additional experiments for assessment of sensorimotor deficit, and novel object location test, as well as other time points for NOR test, to confirm presented results. At the beginning of the subsection 2.3. CMap analysis of POCD signature the detailed explanation of the method is provided while that information is missing for other listed methods. When were the behavioral tests performed? This information should be specified in the section Materials and methods, subsection Behavioral tests for each test. What correlation test was used? The reference for NOR test is not provided where it is indicated, but at it was introduced at the end of the paragraph. In this section, it should be emphasized that the schematic representation of the study is shown in Figure 2A. These are just some examples of the shortcomings of this section. It is necessary to read it thoroughly and rewrite and arranged it to be much more fluent and informative.

Response 2: Thank you for your suggestion. We have accordingly revised the section of Material and Methods in the revised manuscript. 1) The descriptions of the methods have been rearranged in a logical order. 2) The detail information of methods, such as the definition of experimental and control groups, the way of sacrifice of the animals, the type of tissue used, the producers of secondary antibodies, the catalog numbers of the used primers, the time for performing behavioral tests, and the detailed correlation test for Connectivity Map (CMap) analysis, have been provided. 3) Most importantly, to confirm the results in the original manuscript, we have analyzed the data from previous behavioral experiments, and added new figures. Previously, it was reported that the running and swimming speeds could reflect sensorimotor functions of mice [1-3]. In our study, the total distance of mice running were not altered among various groups in the open field tests (Fig. S1A). In addition, the swimming speed of mice were not significantly changed among groups in Morris water maze tests (Fig. S1B). These results suggested that neither acitretin or surgery could induce sensorimotor deficits in animals. We have accordingly added these results in our revised manuscript. However, due to the limited duration for revision, we did not able to perform experiments for novel object location test, as well as time points for novel object recognition (NOR) test.

Fig. S1. (A) The total distance of mice running were not altered among various groups in the open field tests. (B) The swimming speed of mice were not significantly changed among groups in the probe trial of Morris water maze tests. The data were expressed as mean ± SD (n = 6-12 for each group, one-way ANOVA and Tukey’s test).

Fig. S1. (A) The total distance of mice running were not altered among various groups in the open field tests. (B) The swimming speed of mice were not significantly changed among groups in the probe trial of Morris water maze tests. The data were expressed as mean ± SD (n = 6-12 for each group, one-way ANOVA and Tukey’s test).

Comments 3: In section Results the positive and negative correlations are mentioned, but without the correlation coefficient. The information should be provided. The explanation why each method was performed was provided in this section, it is better to transfer those sentences into sections Materials and Methods, or even Discussion.

Response 3: Thank you for your suggestion. We have accordingly provided CMap score, a CMap-specific parameter for evaluating correlation between different signatures, in the revised manuscript. In addition, the explanation why each method was performed was transferred.

Comments 4: Section Discussion is not thorough enough and the key points are not well discussed. The section is written too general without logical connection with the obtained results of this study. For instance, the part of the paragraph is dedicated to PI3-K/AKT signaling, and then the other part is about the drugs for POCD treatment in general. It is unclear whether PI3-K/AKT signaling is altered after the drugs and why is that important for the current study. A very short paragraph about the possible connection between RAR and PI3-K/Akt pathway is provided only at the end of this section, without detailed explanation. Moreover, is there association between RAR and listed treatments, like dexamethasone, memantine, etc?

Response 4: Thank you for your advice. We are sorry for not expressing clearly in our previous manuscript. In this part, we emphasized the negative correlations between postoperative cognitive dysfunction (POCD) and known anti-POCD treatments to demonstrate the reliability of our CMap model of POCD. However, there is no obvious association between retinoic acid receptor (RAR) and these listed treatments. In our revised manuscript, we have rewritten the relevant parts to well discuss the key points.

Comments 5: In the Legend of the Figure 3A the representative WB are presented, not what is indicated below the Figure.

Response 5: We have accordingly revised the legend of figure 3A in our manuscript.

Comments 6: The abbreviations should be defined when first used in the text, and once introduced the abbreviations should be used through the manuscript in the same manner (for instance NOR, CSF, etc). The authors should read the manuscript thoroughly and uniform this item.

Response: We have accordingly revised and uniformed the abbreviations in our manuscript.

Comments 7: It is necessary to pay attention to lower and upper case letters, as well as punctuation marks.

Response 7: We have accordingly revised in our manuscript.

Reference:

  1. Iloun, P.; Abbasnejad, Z.; Janahmadi, M.; Ahmadiani, A.; Ghasemi, R. Investigating the role of P38, JNK and ERK in LPS induced hippocampal insulin resistance and spatial memory impairment: effects of insulin treatment. Excli j 2018, 17, 825-839.
  2. Nwafor, D.C.; Chakraborty, S.; Jun, S.; Brichacek, A.L.; Dransfeld, M.; Gemoets, D.E.; Dakhlallah, D.; Brown, C.M. Disruption of metabolic, sleep, and sensorimotor functional outcomes in a female transgenic mouse model of Alzheimer's disease. Behav Brain Res 2021, 398, 112983.
  3. Khacho, M.; Clark, A.; Svoboda, D.S.; MacLaurin, J.G.; Lagace, D.C.; Park, D.S.; Slack, R.S. Mitochondrial dysfunction underlies cognitive defects as a result of neural stem cell depletion and impaired neurogenesis. Hum Mol Genet 2017, 26, 3327-3341.

Round 2

Reviewer 2 Report

The authors have satisfactorily addressed most of my concerns.